# Reliability and Validity of Six Selected Observational Methods for Risk Assessment of Hand Intensive and Repetitive Work

**DOI:** 10.3390/ijerph20085505

**Published:** 2023-04-13

**Authors:** Teresia Nyman, Ida-Märta Rhén, Peter J. Johansson, Kristina Eliasson, Katarina Kjellberg, Per Lindberg, Xuelong Fan, Mikael Forsman

**Affiliations:** 1Department of Medical Sciences, Occupational and Environmental Medicine, Uppsala University, SE-751 85 Uppsala, Sweden; peter.johansson@medsci.uu.se (P.J.J.); kristina.eliasson@medsci.uu.se (K.E.); 2Department of Occupational and Environmental Medicine, Uppsala University Hospital, SE-751 85 Uppsala, Sweden; 3School of Engineering Sciences in Chemistry, Biotechnology and Health, KTH Royal Institute of Technology, SE-141 57 Huddinge, Sweden; ida.rhen@regionstockholm.se (I.-M.R.); miforsm@kth.se (M.F.); 4Centre for Occupational and Environmental Medicine, Stockholm County Council, SE-113 65 Stockholm, Sweden; katarina.kjellberg@ki.se; 5Unit of Occupational Medicine, Institute of Environmental Medicine (IMM), Karolinska Institutet, SE-171 77 Stockholm, Sweden; xuelong.fan@ki.se; 6Department of Occupational Health Science and Psychology, Faculty of Health and Occupational Studies, University of Gävle, SE-801 76 Gävle, Sweden; per.lindberg@hig.se

**Keywords:** ergonomics, repetitive work, hand intensive, risk assessment, observation, reliability

## Abstract

Risk assessments of hand-intensive and repetitive work are commonly done using observational methods, and it is important that the methods are reliable and valid. However, comparisons of the reliability and validity of methods are hampered by differences in studies, e.g., regarding the background and competence of the observers, the complexity of the observed work tasks and the statistical methodology. The purpose of the present study was to evaluate six risk assessment methods, concerning inter- and intra-observer reliability and concurrent validity, using the same methodological design and statistical parameters in the analyses. Twelve experienced ergonomists were recruited to perform risk assessments of ten video-recorded work tasks twice, and consensus assessments for the concurrent validity were carried out by three experts. All methods’ total-risk linearly weighted kappa values for inter-observer reliability (when all tasks were set to the same duration) were lower than 0.5 (0.15–0.45). Moreover, the concurrent validity values were in the same range with regards to total-risk linearly weighted kappa (0.31–0.54). Although these levels are often considered as being fair to substantial, they denote agreements lower than 50% when the expected agreement by chance has been compensated for. Hence, the risk of misclassification is substantial. The intra-observer reliability was only somewhat higher (0.16–0.58). Regarding the methods ART (Assessment of repetitive tasks of the upper limbs) and HARM (Hand Arm Risk Assessment Method), it is worth noting that the work task duration has a high impact in the risk level calculation, which needs to be taken into account in studies of reliability. This study indicates that when experienced ergonomists use systematic methods, the reliability is low. As seen in other studies, especially assessments of hand/wrist postures were difficult to rate. In light of these results, complementing observational risk assessments with technical methods should be considered, especially when evaluating the effects of ergonomic interventions.

## 1. Introduction

Work-related musculoskeletal disorders (WRMSD) are still a major concern in working life [1]. These disorders can, besides pain and suffering for the individual, cause the employer economic consequences due to sick leave costs and reduced productivity [1,2,3,4,5]. Physical factors in the work environment, such as forceful exertions, awkward postures and repetitive work, as well as psychosocial and organisational factors, are associated with WRMSDs in the neck, shoulder and arms [6,7,8,9,10,11,12]. Hence, risk assessments of physical factors are of importance for identifying potentially harmful work tasks and for prioritizing and designing workplace interventions, both regarding the physical design of the workplace, work technique and work organization [13,14]. For evaluation purposes, it is also recommended to perform renewed risk assessments after an intervention. Furthermore, governmental bodies, such as national or regional work environment authorities, stipulate that risk assessments should be conducted within the systematic occupational health management work [15]. It is therefore highly important that methods for risk assessments are valid and reliable.

Through ergonomics research in combination with technical developments, it has become less complicated and less expensive to perform risk assessments through direct measurements using different types of technical equipment, and several of these methods are now becoming available also for ergonomists within the occupational health services (OHS), or equivalent [16,17]. However, the most common way for practitioners to identify and quantify physical exposures at a workplace is still to use observations. Previous research indicates that ergonomists often assess risks in the work environment solely by means of observation, based on his/her knowledge and experience, without the use of any systematic methodology or explicit method [18,19,20]. Further, Eliasson et al., concluded that the reliability of such assessments is low [21]. To improve the systematics of observations, different observational-based methods can be used. These methods are described as being useful due to their low cost and ability to present the results of the risk assessment in a way that is easy to understand for the stakeholders [18,22].

Different observational methods are designed to target and assess different exposures, such as manual handling, (heavy lifting, push-pull actions), awkward postures or repetitive work/hand intensive work. The ergonomist often needs to combine several observational methods when performing a comprehensive risk assessment of a workplace. In a review article by Takala et al., 2010, 30 eligible observational methods were identified, and it was concluded that for many methods studies of reliability and validity were lacking [22]. Since then, a number of reliability studies of different observational methods have been conducted, but the studies show mixed results (see Appendix A; Table A1 and Table A2). A recent systematic review by Graben et al. concluded that comparisons of reliability and validity between different methods are hampered by differences in the study design between existing studies, e.g., regarding the background and competence of the observers, the complexity of the observed work tasks, whether the observations are done directly or digitally, as well as the differences in the chosen statistical methodology [23].

For the assessment of hand intensive and repetitive work, some of the more well-known and more commonly used methods are the Assessment of repetitive tasks of the upper limbs (ART) [24], the Hand Arm Risk Assessment Method (HARM) [25], the Occupational Repetitive Actions Checklist (OCRA) [26], the Quick Exposure Check (QEC) [27] and the Strain Index (SI) [28]. These methods have all been shown to have a reasonably good predictive ability, either in direct studies, or indirectly, for the methods that are based on another method or on epidemiologically documented risk factors [22,25,26,27,28,29].

Furthermore, the methods have all been evaluated with regards to their inter-observer reliability, with the reliability commonly interpreted as fair, but rarely above a moderate level [27,30,31,32,33,34,35,36,37,38,39]. Moreover, these methods (with the exception of HARM) have been evaluated concerning intra-observer reliability, which usually has been proven to be higher [27,30,31,34,35,37,40]. For further information regarding these studies, see Appendix A (Table A1 and Table A2).

To conclude, there are large methodological differences between studies. The occupations and work tasks where the methods have been evaluated differ, as well as the statistical methods used for evaluation of the reliability. Furthermore, the observers’ background regarding ergonomics expertise varies between studies, such as the observers being workers with no prior education within ergonomics, or occupational health students or experienced ergonomists within occupational health service. All these factors can influence the results in the various studies and makes a comparison between the studies and the evaluated methods hard to perform. It is therefore of interest to include several methods in the same study and assess the reliability and validity of these methods using the same study design, taking all the aforesaid factors into consideration. 

The purpose of this project was to evaluate the above-mentioned five risk assessment methods, concerning both inter- and intra-observer reliability as well as concurrent validity, using the same observers, the same occupations and work tasks and the same statistical methods. 

Additionally, since this project was performed in a Swedish context, a sixth method, the most commonly used among ergonomists within the OHS in Sweden, the Repetitive work and work posture models by the Swedish Work Environment Authority (SWEA) was included [18].

## 2. Materials and Methods

Inter- and intra-observer reliability of six different observational methods were assessed by letting twelve experienced ergonomists perform risk assessments of ten video-recorded work tasks twice. Concurrent validity was assessed by comparing the ergonomists’ assessments with consensus assessments carried out by three experts within ergonomics.

### 2.1. Included Methods

All methods except OCRA were already translated into Swedish prior to the present project and were, prior to and at the time of the project publicly available, either through the Swedish Work Environment Authority’s website, or through the website hosted by the Department of Occupational and Environmental Medicine, Uppsala University Hospital. The OCRA method was translated into Swedish by the authors of the present study with support from Professor Daniela Colombini [34]. Several of the methods are well-known by Swedish ergonomists in occupational health service [18], and training in these methods are often a part of the continuing education courses and university level programmes in ergonomics in Sweden. The six methods are described below. For details on previous reliability studies regarding the six methods, see Appendix A (Table A1 and Table A2).

1. Assessment of repetitive tasks of the upper limbs (ART) was developed by the British Health and Safety Executive (HSE) [24]. When using the method, an assessment is made in four different areas: frequency/repetition of movements, force demands, work postures (neck, back, arm/shoulder, hand) and other factors (including work pace and task duration). The calculated score is translated into one of three risk levels: green, yellow or red (corresponding to low, moderate and high risk). Previous studies of the inter-observer reliability of the final risk score have shown Intra class correlations (ICCs) ranging from 0.73 to 0.87, which can be interpreted as moderate to good reliability. Intra-observer reliability studies have shown that ICCs for the final risk score and overall risk level were ranging from 0.84 to 0.99 and 0.90, respectively, indicating a good to excellent reliability [41,42].

2. Hand Arm Risk Assessment Method (HARM) was developed by the Netherlands Organisation for applied scientific research (TNO) [25]. When using the method, an assessment is made in five different areas: task duration, force demands (including frequency/repetition of grasp), work postures (neck, arm/shoulder and forearm/wrist), exposure to vibrating tools and other factors (such as precision demands, adverse climate and pauses). The calculated score is translated into one of three risk levels: green, yellow or red (corresponding to low, moderate and high risk). The only previous study found of the inter-observer reliability indicated a moderate to good reliability, with an ICC of the final risk score of 0.73. No studies of the intra-observer reliability were found.

3. Occupational Repetitive Actions of the Upper Limbs checklist (OCRA) was developed in Italy and is a simplified version of the OCRA index [26,43,44]. When using the checklist, assessments are made in six different areas: work postures, frequency/repetition of movements, (arm/shoulder, elbow, wrist and hand), force demands, task duration, lack of recovery time and other factors (physio-mechanical as well as socio-organisational). A calculated risk score is translated into one of five risk levels: green, yellow, light red, dark red and purple (corresponding to acceptable, very low, medium-low, medium and high risk). Previous studies have shown ICCs of the overall risk levels ranging from 0.62 to 0.80 for the inter-observer reliability [31,33], and 0.85 for the intra-observer reliability, indicating a moderate to good reliability and a good reliability, respectively [31].

4. Quick exposure check (QEC) was developed in the UK [27,45]. When using the method, an assessment is made in six different areas: work postures, frequency/repetition of movements, (back, neck, arm/shoulder, wrist/hand), force demands, task duration, exposure to vibrating tools and other factors (such as visual demands, work pace and stress). The calculated score is translated into one of four risk levels: low, moderate, high and very high risk. The risk levels are presented specifically for back, neck, arm/shoulder and wrist/hand. Additionally, a total score has been suggested by Brown and Li, and is also used in the present study [46]. Previous studies of the inter-observer reliability for the total score have revealed ICCs ranging from 0.71 to 0.97 [35,47], considered as moderate to excellent reliability, and ICCs of the intra-observer reliability between 0.4 and 0.89 [35,37], which indicated a poor to good reliability. 

5. Strain Index (SI) was developed in the United States as a method for analysing jobs with a risk of distal upper extremity disorders [28]. When using the method an assessment is made in six different areas: intensity of exertion, duration of exertion per work cycle, efforts per minute, wrist posture, speed of exertion and task duration. The calculated score is translated into one of three risk levels: low, moderate and high risk [39]. Since the development of SI, an updated version has been published [48]. However, at the time of data collection in the present study, the version was not yet available. Previous inter-observer reliability studies have shown that the ICC for the risk level was 0.54 and for the risk score was between 0.43 and 0.64 [31,33,38] indicating moderate reliability as well as poor to moderate reliability, respectively, while studies of the intra-observer reliability have shown ICCs for the risk level ranging from 0.56 to 0.82 and an ICC for the risk score of 0.76, indicating moderate to good reliability as well as moderate reliability, respectively [31,40].

6. Repetitive work and work posture models (SWEA) is a two-part checklist included in the Swedish Work Environment Authority’s (SWEA) provisions on physical ergonomics, AFS 2012:2, and was originally developed in a pan-Nordic project in 1994 [49]. When using the checklist for repetitive work, an assessment is made in four different areas: work cycle, postures and movements, scope for action and work content. The classification for each of the areas is made in three levels: green, yellow or red (corresponding to low, average and high). Regarding a number of aggravating factors, the time when the work is performed and how it is distributed over the day, the assessor makes a summary assessment of the included parameters, whereby the work cycle is considered the overriding factor. In the work postures checklist, an assessment is made separately for neck, shoulder/arm, back and leg. The classification for each of the areas is made in three levels: green, yellow or red (corresponding to low, moderate and high risk). No previous studies of the inter- and intra-observer reliability have been found.

### 2.2. Recruitment

Twelve ergonomists, all registered physiotherapists (RPT) (a common combination in Sweden), were recruited through contacts with different OHS companies or through social media posts to members of the Swedish Ergonomist and Human Factors Society (EHSS). To be included in the study, the ergonomists should be employed by an OHS company (or equivalent) and have at least one year of work experience with risk assessments. All twelve ergonomists were women, mean age 49 years (range 39–55 years). All had extensive work experience within physical ergonomics, mean 13 years (range 4–26 years), and ten out of twelve ergonomists performed more extensive risk assessment assignments four times per year or more often. Prior to the study, all ergonomists had used the SWEA method, six of the ergonomists had used HARM and five ergonomists had used QEC. Only one ergonomist had used SI and none of the ergonomists had used ART or OCRA. 

### 2.3. Training of Ergonomists

Initially, the ergonomists individually learned and trained on the six selected methods during a three-week period using an e-learning platform [50]. The training encompassed: (1) a pre-recorded lecture on general aspects of risk assessment, (2) pre-recorded instruction videos with “walk-through” examples where the six selected methods were applied on different work tasks and (3) self-supported training using a video library with film clips (two to six minutes long) of different work tasks. The films were accompanied by written information on the work task, e.g., task duration, pause and rests, weights of handled goods, and employees’ ratings of force exertion, discomfort and work demands. The manuals and the protocols for all six methods were available for download on the e-learning platform. 

### 2.4. Risk Assessments

Ten video-recorded work tasks from different work sectors were included (See Table 1). For all chosen work tasks, the work postures and movements were largely of a repetitive character (Table 1).

The video recordings were made using two to four video cameras from different angles to enable the best possible conditions for the risk assessments. For each work task, the different views were synchronized into one video with multiple windows to show the different views of the worker with a close-up on hand and wrist movements. Each of the finalized video recordings was two to six minutes long.

During the performance of the risk assessments the ergonomists, seated in the same room, watched the video recordings on individual laptop computers and were allowed to pause or repeat the playback as needed. The ergonomists were requested to perform the assessments individually, without conferring with each other.

Not all items included in the methods could be rated by solely looking at the video. Consequently, additional information was pre-given to the ergonomists in a supplementary document. This document covered information such as the duration of work tasks during the work day, pause and rest schedules, weight of handled goods, visual demands, worker’s ratings of discomfort on Borg’s CR10-scale [51] as well as the level of work demands and control (Table 1). To enable intra-observer reliability analyses, the risk assessments performed were repeated in a second session, with at least four weeks between the occasions [52]. 

### 2.5. Analyses of Reliability and Validity

Calculations of inter-observer reliability were based on the ergonomists’ assessments in the first session. Calculation of intra-observer reliability was based on the first and second session for the ergonomists that repeated their assessments. Statistics for each of the ten work tasks regarding the individual items rated in the methods, as well as for the total risk scores of each method, were calculated. The risk scores were then transposed into risk levels according to the specific instructions for each of the methods. 

For each method, except for SWEA, the duration of the work task is rated and given a value which influences the final score. In the instructions to the ergonomists, information regarding the duration of each work task was pre-given in a supplemented document (Table 1). To decrease the variability between the work tasks, inter-observer reliability was also calculated imputing a standardized work task duration, using 3 h 45 min for all ten work tasks.

Further, the six methods differed somewhat with regards to their coverage of exposure variables/body regions, see Table 2.

In the present study, the methods’ concurrent validity was evaluated. Consensus assessments were carried out by three experts, each expert with more than 20 years’ experience in both performing risk assessments of physical exposures, as well as university level teaching within this field. The experts also had extensive research experience, as well as governmental experience concerning work-environment legislation [53].

In the first step, the three experts made individual assessments in accordance with the manual of each method. In the second step, the three experts compared and discussed their individual assessments until consensus was reached. In the third and final step three months later, they (together in the group) repeated the assessments in the reversed order of methods. The assessments from this time were very similar to those of the first time. They decided upon the few discrepancies and agreed upon their final consensus assessments. The experts’ consensus assessments were used as a gold standard in the computation of the concurrent validity of the ergonomists’ ratings.

### 2.6. Statistical Methods

The proportional agreement in percent and the linearly weighted Kappa coefficient (K_lw_, see below) were chosen as the primary parameters for the analyses of both inter- and intra-observer reliability as well as for concurrent validity. However, to enable comparisons with other studies, several additional parameters were computed. 

Proportional agreement (%) was calculated as the number of rating pairs in agreement divided by the total number of rating pairs. However, to take agreement due to chance into account, proportional agreement should be presented together with other parameters, for example, kappa statistics [52,54]. Cohen’s kappa was calculated for both inter-observer and intra-observer reliability; for intra-observer reliability, the kappa value was calculated for each of the observers, and then the mean value of these kappa values was calculated [54]. Similarly, for the concurrent validity, the kappa value was calculated for each of the ergonomists paired with the experts’ consensus assessments, and then the mean value of these kappa values was calculated. For the inter-observer reliability, pairwise kappa values were first computed, and then averaged over all pairs, since Cohen’s kappa is only applicable for when two observers are used or when test–retest reliability is evaluated. This averaging was conducted in the way suggested by Davies and Fleiss (1982), where the expected agreement, P_e_, in Cohen’s kappa formula for each pairwise comparison, *K* = (P_o_ − P_e_)/(1 − P_e_), is substituted with the average P_e_ of all pairs [55]. P_o_ is the proportional agreement. 

However, Cohen’s unweighted kappa does not distinguish minor from major discrepancies in ratings, and since the risk ratings for the different included methods represent ordinal data (low, moderate, high risk, very high risk, etc.), linearly weighted kappa [56,57] was computed and averaged in the same way as the unweighted kappa [55,58,59].

The intraclass correlation (ICC) two-way absolute agreement method 2.1 was used in accordance with Shrout and Fleiss [60], and was computed to facilitate comparisons with other studies [27,33,35,39,40]. ICC is mostly applicable for continuous data but has been used in previous reliability studies on ordinal data. Kendall’s coefficient of concordance (KCC) was also computed. KCC is a non-parametric relative to ICC that is applicable with ordinal data [61].

For the interpretation of kappa values, the recommendations (fair 0.21–0.40, moderate 0.41–0.60, substantial 0.61–0.80, almost perfect 0.81–1.00 and perfect 1.00) by Landis and Koch was used [62]. 

In cases where an observer had missed to fill in a rating for an item, the missing value was replaced by the median category of the other observers’ ratings. See Table 3 for the percentages of these occurrences. 

The statistical computations were carried out using scripts written in MATLAB version 8.5 (MathWorks Inc., Natick, MA, USA), the output parameters of which, for small samples, were compared and found to agree with corresponding parameters of the statistical software R version 4.1.3 (R Foundation for Statistical Computing: Vienna, Austria, 2022) or SPSS version 27 (IBM: Armonk, New York, NY, USA, 2022). MATLAB was used in order to obtain time-effective analyses, since there were no functions for multi-observer linearly weighted kappa in R or SPSS.

### 2.7. Ethical Considerations

The Regional Ethical Review Board in Stockholm (Dnr 2013/308–31/3) gave ethical approval for the present study. Informed oral and written consent was obtained from both the ergonomists performing the risk assessments as well as from the workers featured in the video-recorded work tasks. Ethical considerations of relevance may be the video recordings, which could be considered an intrusion of privacy for the participating workers. For the ergonomists, the results of their individual risk assessments may constitute a transparency risk. Hence, both the workers and the ergonomists were informed that the data was constricted to the project group members, and that any results from the study would be presented only on group level.

## 3. Results

Twelve ergonomists were recruited for this project. However, not all of the ergonomists completed all risk assessments with all methods.

In the first session, at least 10 out of 12 recruited ergonomists completed the assessments of all items in all methods (Table 3). In total, 9168 item ratings were performed, and missing ratings (1.4%) were replaced with the group median value. Based on the item ratings, altogether, 1230 overall risk level assessments were made. 

In the second session, at least 6 out of 12 recruited ergonomists completed the assessments using all methods (Table 3). In total, 7093 item ratings were performed, and missing ratings (0.9%) were replaced with the group median value. Based on the item ratings, altogether, 900 overall risk level assessments were made. 

The expert group performed in all 2520 item ratings, making 90 overall risk level assessments. The expert group did not perform any risk assessments for SWEA neck posture, SWEA shoulder/arm posture or SWEA back posture. 

The distribution of the risk levels assessments from the ergonomists first and second session as well as the expert groups’ ratings (ranging from low risk level to high risk level as stipulated in each method) is shown in Table 3.

### 3.1. Reliability

#### 3.1.1. Inter-Observer Reliability

The averaged linearly weighted kappa (K_lw_) of the overall risk level assessments differed between the six methods, showing the highest inter-observer reliability for HARM and ART (0.65), and the lowest for SWEA overall postures and movements (0.21) (Table 4). When the standardized work task duration was imputed to all work tasks, the linearly weighted kappa (K_lw_) decreased for all methods except OCRA, and the decrease was largest for HARM with a decrease from 0.65 to 0.26 (Table 5).

Further, it was of particular interest to analyse more in detail the items that the ergonomists rated by actually observing the worker in the video-recorded work tasks, such as postures and movement/repetition (Table 2). The results of the linearly weighted kappa (K_lw_) show that the inter-observer reliability, especially for hand/wrist posture, was very low for several of the methods, and that items relating to movements, such as repetition, showed the highest inter-observer reliability (Table 6). Complete results for all rated items per method are shown in Appendix B (Table A3).

The ergonomists obtained the same risk level assessments to different degrees in the ten work tasks. In Figure 1 an example is given from the first and second session using the risk level assessments for QEC (Figure 1). The figure shows that, concerning the inter-observer agreement, the risk level assessment was the same for all 10 ergonomists in some of the work tasks, whilst for other work tasks the risk levels differed.

#### 3.1.2. Intra-Observer Reliability

The intra-observer reliability of risk level averaged over the ten work tasks are presented in Table 7. However, the level of intra-observer agreement differed between work tasks for all methods. In Figure 1 an example is given from the first and second session using the risk level assessments for QEC. 

The linearly weighted Kappa (K_lw_) for the overall risk level assessments differed between the methods, ranging between 0.30 to 0.70, showing the highest agreement for HARM, and the lowest for SWEA postures and movements. When the standardized work task duration was imputed to all work tasks, the linearly weighted kappa (K_lw_) decreased for all methods, and the decrease was largest for HARM with a decrease from 0.70 to 0.47 and ART (left arm a decrease from 0.65 to 0.36, and right arm from 0.68 to 0.43) (Table 8).

### 3.2. Validity

The concurrent validity of the overall risk level assessments was assessed by comparing the ergonomists’ ratings to the ratings made by the expert group (Table 9). In these computations, the standardized work task duration was imputed to all work tasks. The validity differed between the methods, with the highest linearly weighted kappa averaged (K_lw_) for HARM (0.42) and ART (0.54), and the lowest K_lw_ for SWEA overall repetition (0.31). Regarding QEC, besides the method giving a total risk level (K_lw_ = 0.47), risk level assessments were also computed for individual body parts (neck, shoulders, wrist and back), showing the highest K_lw_ for neck (0.88), and the lowest K_lw_ for back (0.31).

## 4. Discussion

The objective of the present study was to assess the inter-observer reliability, the intra-observer reliability, as well as the concurrent validity, of six well-known methods that previously have been presented as suitable for assessing risk in work tasks that are physically demanding for the upper extremities. The work tasks chosen for the risk assessments displayed various levels of exposure to hand-intensive and repetitive work tasks, as well as different degrees of complexity. Previous studies on the reliability of observational risk assessment methods have been performed on different materials and have used different reliability measures in the evaluations (See Appendix A, Table A1 and Table A2). Therefore, a second objective in the present study was to include several methods in the same study and assess the reliability and validity of these methods using the same study protocol.

### 4.1. Inter-Observer Reliability

The results show that for all methods (overall risk levels) the inter-observer percentage agreement ranged between 39 to 83, showing the highest agreement for SI and the lowest agreement for OCRA. The linearly weighted Kappa (K_lw_) for the inter-observer reliability differed between the methods, ranging from 0.18 (SI) to 0.65 (HARM and ART). As for the concurrent validity, the results were comparable to the inter-observer reliability. According to the criterions suggested by Landis and Koch [62], a Kappa value between 0.60 and 0.80 can be considered to represent “substantial agreement”, whilst a Kappa value between 0.21 and 0.40 equals a “fair agreement”. The majority of the Kappa values were found in the range between 0.41 and 0.60, which represents a “moderate agreement”. However, these thresholds are indicative and arbitrary, and they usually refer to unweighted kappa, which are lower than the linearly weighted kappa used here. Therefore, the thresholds should be interpreted with precaution. If the outcome and consequences of the assessments had been more critical, higher criteria should have been used. It is also possible to argue that stricter criteria should be applied to intra-observer reliability than to inter-observer reliability because the between-observer variance is not present in intra-observer reliability. However, for simplicity we chose the same criteria for both intra- and inter-observer reliability.

Compared to other reliability studies of the six included methods in this study, our results showed both higher and lower inter-observer reliability. 

For example, as for the inter-observer reliability of the QEC method, the ICCs for the total score have been shown to be higher (0.71–0.87 [47], 0.86 [35] and 0.93 [37]), than what we found (0.69), while others, using different kappa statistics, have reported values for each body part and item assessed (K_lw_ = 0.61–0.85 [36]), that were close to ours (K_lw_ 0.35–0.85) or considerably lower (K = 0–0.47 [27]) than our results (K = 0.24–0.82).

Regarding the SI method and the inter-observer reliability of the total risk score, a study of cheese production tasks [33], similar in design to our study, showed a substantially higher ICC (0.59) compared to ours (0.18). This was also the case when comparing to other reliability studies of the SI, such as studies of tasks within poultry slaughtering (ICC = 0.54 [31]) as well in health care and manufacturing, the latter using linearly weighted kappa statistics (K_lw_ = 0.41 [39] compared to our result K_lw_ = 0.18). Higher ICC compared to ours have also been shown in a study of manufacturing and material handling [38], where the ICC for individual assessments was lower (0.43) than for assessments made in teams (0.64). 

Our results of the ART method showed ICCs for the total risk level (0.70–0.75) that were lower than previously found (0.75, 0.87 [30] and 0.77 [31]). Likewise, our results of the ICCs for the total risk level of the OCRA-method were lower than described by others (0.80 [33] and 0.72 [31]).

The only study found of the inter-observer reliability of the HARM method [32] included similar assessment tasks as in our study. The study showed an ICC for the total risk level (0.73) which was lower, but close, to our result (0.77).

The inter-observer reliability of SWEA-method in our study was found to be one of the lowest among the included methods, with kappa values indicating a fair agreement. No other studies, known to us, have previously assessed the inter-observer reliability of this method.

### 4.2. Intra-Observer Reliability

As expected, a somewhat higher reliability was found within observers (intra-observer reliability) compared to between observers (inter-observer reliability), with an overall percentage agreement ranging between 45% in OCRA to 79% in ART. The corresponding K_lw_ ranged between 0.13 for SI and for SWEA (Shoulder/arm posture) to 0.88 for QEC (Neck). Compared to other studies of the six methods included in our study, our results of the intra-observer reliability were generally lower. 

Previous studies of the QEC method have shown both lower and higher intra-observer reliability compared to what we found. Regarding the ICC for the total risk score, our ICC was higher (0.79) than in the study of five observers and 13 tasks (0.41–0.60 [35]) but lower than shown in a study of one observer and one task (0.89 [37]). In studies where reliability statistics only have been reported for each body part and items, the ICC values of hospital cleaning tasks (0.61–1.00 [63]) and kappa values of industrial tasks (0.45–0.53 [27]) have shown to be higher than what we found (ICC = 0.1–0.72 and K= 0.32–0.59, respectively).

Our findings of the ART method showed that the ICCs for the total risk level (0.74, 0.78) were lower than previously reported (0.84, 0.99 [30] and 0.9 [31]). Even our ICC for the risk level of the OCRA method (0.72) was lower to what has been reported by others (0.85) [31]. 

Concerning the SI method, our study showed substantially lower ICCs for the total risk level (0.10) to a study of manufacturing tasks [40], both for individual observers (0.56), as well as compared to the results when the same tasks were assessed in teams (0.82). 

In the present study, we found that the intra-observer reliability of HARM was the highest among the methods with a kappa value that could be interpreted as substantial agreement, while SI and SWEA showed the lowest reliability and kappa values that indicate a fair to moderate agreement. No previous studies, as far as we have found, have examined the intra-observer reliability of either the HARM or the SWEA method.

### 4.3. Further Discussions of the Results in the Present Study

A tentative explanation as to why the HARM and ART methods showed the highest inter-observer agreement (K_lw_), were at first thought to be the layout design of the scoring sheets, which for both methods included comprehensible photographs/drawings showing neutral and awkward postures for different body regions such as wrist, elbow, arm and neck. However, when studying the computed separate K_lw_ for each rated item, all K_lw_ for items concerning postures and repetition/movements were below 0.44, which is comparable to the findings in a previous study by Eliasson et al. (2017) where ergonomists performed risk assessments without the use of any specific method nor photographs/drawings [21]. Hence, the well-defined illustrations did not seem to facilitate the ratings of postures and movements. The relatively high inter-observer reliability of HARM and ART instead seems to be explained by the supplemented information concerning each work task that was pre-given to the ergonomists (see methods section). Task duration especially has a high impact in these two methods on the resulting estimated risk level; therefore, the agreement in risk levels was high when different times were assigned different work tasks (Table 4), but low when a standardised time was used (Table 5). This raises thoughts on how the results would have turned out if the ergonomists had not had any pre-given information, but had had to get this by individual interviews of the workers, which normally is the case. Throughout the methods, the K_lw_ was generally the lowest for the item ratings of postures of small body regions such as the wrist and hand. This is in line with previous research, which has identified the challenges of visually assessing postures and movements of small body regions [22,64,65,66].

In the present study, we have chosen to use both the proportional agreement (%) and the linearly weighted kappa (K_lw_). These two parameters do not always correspond, and the reason why it is important to include the proportional agreement (%) in the result is because there are situations where the linearly weighted kappa (K_lw_) may be low whilst the proportional agreement (%) is high. As an example, if the observers were to choose between standing and sitting and almost everyone choose standing, there would be a high agreement (%), but also a high expected agreement, P_e_, and a low K_lw_ (see formula in Section 2.6.) [67]. This explains why the K_lw_ for SI regarding intra-observer reliability is only 0.13 while the proportional agreement is 77%. This may also be an explanation for the intra-observer reliability results regarding SWEA (Shoulder posture).

Regarding the validity, the K_lw_ were in line with the inter-observer reliability for each method, which can be expected since the validity parameter calculated in the present study was concurrent validity. Where there is a large inter-observer variation, as for instance in wrist posture ratings, there is also a large variation around the gold standard ratings. Similarly, for a method with a large inter-observer variation in the total score, there are also large variations when the observer’s scores are compared to the gold standard total score.

As shown above, and in agreement with previous findings, there is a considerable variation between ergonomists’ assessments of risk levels for MSDs in the observation methods. However, since observation without the use of any specific method have a lower and non-acceptable reliability [21], it is recommended to use one or more systematic observational-based risk assessment methods. Another approach would be to combine an observational method with validated methods of direct measurements where the items of the lowest reliability in the observational method of choice are replaced by using technical methods, especially so when an intervention is to be evaluated.

Regardless of the choice of method used for quantification of exposure levels, it is important to consider that additional information regarding, e.g., psychosocial and organizational factors, as well as results from occupational health surveillance regarding musculoskeletal disorders, is needed for a more comprehensive risk assessment [68,69]. Furthermore, the use of observational risk assessment methods includes many different aspects in a compact way. Several methods also involve the experiences and opinions of the worker in the assessment. The methods may facilitate by increasing the interest of work environment and ergonomics, the knowledge of different risk factors, and may provide a basis for a participatory approach in the risk assessment. 

The inter- and intra-observer reliability in the present study may have been affected by several factors. Although the video recordings were short, the observers may still have concentrated on different parts of the recordings, which may lead to differences in the assessments. Further, the previous knowledge and experience of risk assessment in the different work sectors may have differed between the observers. 

A further affecting factor could be difficulties in assessing postures and movements from video recordings compared to live observations. However, Mathiassen et al. (2013) found that posture assessments based on video recordings is beneficial, since this provides the possibility to conduct repeated ratings of the same work sequence [70]. In the present study, the video recordings were composed of synchronized video windows that displayed the work sequence from several different angles, something which may have contributed to provide a more comprehensive picture [40,71].

Regarding the intra-observer reliability, an often stated problem is possible changes between the test and the re-test occasion [72]. In the present study, this problem is addressed by a design where the assessments are made using video recordings, meaning that the potential sources of variation (such as alteration in job performance or change of the worker performing the job) is overcome.

### 4.4. Methodological Considerations

The data in the present study consists to a large extent of variables with more than two scale steps. Hence, linearly weighted kappa was considered to be the most suitable choice for the analyses of both inter- and intra-observer reliability as well as for concurrent validity. Linearly weighted kappa can differ between one- and two-step differences (a two-step difference is given double the weight of a one-step difference), which is not the case with Cohen’s unweighted Kappa or quadratic weighted kappa [56,57,73]. While the unweighted kappa and agreement percentage become lower with an increased number of items, the linearly weighted kappa is not dependent on item numbers, and facilitates comparisons of assessment methods with different numbers of steps in their scales [56]. 

However, to be able to compare the findings in the present study with those from earlier studies on reliability and validity of observational methods (see Appendix A; Table A1 and Table A2), the choice was made to include several of the most commonly used statistical parameters such as the Cohen’s unweighted Kappa, the intraclass correlation coefficient (ICC) and Kendall’s coefficient of concordance (KCC) [60,61]. The intraclass correlation coefficient (ICC) is often recommended for use in multi-observer comparisons and KCC is a non-parametric relative to ICC. In theory, if all ergonomists perform the assessments with perfect agreement, the validity may still be low (i.e., if all observers do the same but incorrect assessments). On the contrary, when the reliability is low (low agreement) there cannot be a high validity, given that a high reliability is a necessary but not sufficient condition for high validity [74].

The present study investigated the concurrent validity, with the expert group as gold standard. Another approach would have been to use technical measurements of postures and movements as gold standard. However, this was not within the scope of the present study.

As for other types of validity, such as predictive validity, observational methods have been sparsely evaluated. However, the SI method has previously been found to be associated with disorders in the distal part of the upper extremities [39]. In the present study it was not possible to investigate the predictive validity, for which longitudinal data and large cohorts would have been required [75]. 

In the present study, all ergonomists were used to performing risk assessments with different observational methods in their work, and they underwent the same training in the six included methods. The training was aimed at ensuring that the ergonomists had equal minimum level required knowledge of each of the methods. The ergonomists also had access to the training material during the whole period. However, it cannot be ruled out that the differences regarding their prior knowledge and experience in the six methods might influence the reliability scores. The SWEA method was familiar to all ergonomists beforehand, whilst the methods OCRA and ART had not been used by any of the ergonomists prior to the present study. However, if previous familiarity of a method would have affected the results, it is plausible to assume that the reliability scores for SWEA would have been higher, especially with regards to intra-observer reliability, which was not the case in the present study. 

### 4.5. Strengths and Limitations

The heterogeneity of a studied phenomenon (in the present study equal to the work tasks) is an important parameter in reliability studies [76]. Even though the selection of the ten work tasks represented varying degrees of repetitiveness and risk levels, a higher number of diverse work tasks would have improved the study, and there are still many occupations and work tasks that were not included. Nevertheless, because of the large variation between work sectors and job characteristics between the work tasks in the study, the results can be considered to represent repetitive jobs rather well. Further, to ask the participating ergonomists to perform even more ratings than what was demanded in the present study was considered unjustifiable. Still, since the ergonomists agreed to different degrees in the various work tasks, the results may have been somewhat different if other or more tasks had been included.

A limitation in the present study is that the number of ergonomists that completed all assessments decreased, for various reasons, during the data collection period. However, it can be seen as a strength that the inter-observer reliability computations, as well as the validity computations, are based on at least eleven ergonomists, and that the intra-observer reliability computations are based on at least six ergonomists (for most of the computations, eight to ten ergonomists). It is not uncommon that reliability computations are based on fewer observers [27,33,35,39]. Further, all ergonomists were experienced within their occupation, had previous experience of risk assessment assignments and performed risk assessments regularly. A possibility in the present study would have been to also ask the ergonomists to provide subjective ratings of the different methods with regards to usability and feasibility, in relation to observation of different work tasks (postures, movements and body parts). However, this was outside of the scope of the present study.

The group of observers in the present study consisted of solely women, and any systematic gender difference in the assessments could not be investigated, meaning that if there would be a gender dependence where women constantly rate items in the methods higher or lower, it would have influenced the reliability parameters in a negative way. However, the observers represent rather well the population of ergonomists in Sweden, where approximately 80% of the ergonomists with a professional background within physiotherapy are women [34].

Normally, in real life, the ergonomist meets the individual worker at the workplace and hence has the opportunity to investigate the workplace in more detail and to question the worker concerning additional information that adds to the risk assessment. Moreover, the ergonomist is then often able to observe the worker from different angles and for a longer period of time, which gives a more comprehensive representation of the work postures and movements. In the present study, the ergonomists performed their risk assessments by watching videos of workers performing different work tasks, and the rather short length of the videos may have influenced the possibilities to observe postures and movements. Moreover, any interaction with the worker/workplace was not possible. To manage this limitation in the present study, the ergonomists were supplied with written work environment-related information concerning the exposures in different work tasks. Hence, all ergonomists received this very same information; if they had collected this information themselves as in real life, and then used the methods, the reliability would have been lower. 

### 4.6. Future Research

The scope of the present study was to investigate the reliability of several risk assessment methods using the same material and calculating the same statistical parameters. Few studies have compared to what extent different methods agree as to the risk levels calculated [46,77,78]. Further, a review article by Joshi et al., 2019, comparing different observational methods, shows that the result of a risk assessment is dependent on what method is used, and that the correlation of outcome between different methods is weak [79]. There is still a need for more such studies, with several methods and the same material. 

Further, based on the results in the present study concerning the low reliability of assessments of work postures and movements, especially regarding the wrist and hand, future research should focus on developing methods that can combine the accuracy of direct measurements with existing observational methods.

## 5. Conclusions

All methods’ total-risk linearly weighted kappa values (when all tasks were set to the same duration) were lower than 0.5 (0.15–0.45). Moreover, the concurrent validity values were in the same range with regards to total-risk linearly weighted kappa (0.31–0.54) Although these values are often considered as being fair to substantial, they mean agreements lower than 50% when the expected agreement by chance has been compensated for. Hence, the risk of misclassification is substantial. The intra-observer reliability was only somewhat higher (0.16–0.58). Regarding the ART and HARM methods, it is worth noting that the work task duration has a high impact in the risk level calculation, which should be considered in studies of reliability. This study indicates that when experienced ergonomists use observational risk assessment methods, the reliability is low. As seen in other studies, assessments of hand/wrist postures were especially difficult to rate. In light of these results, complementing observational risk assessments with technical methods may be needed, especially when evaluating the effects of ergonomics interventions.

## Figures and Tables

**Figure 1 ijerph-20-05505-f001:**
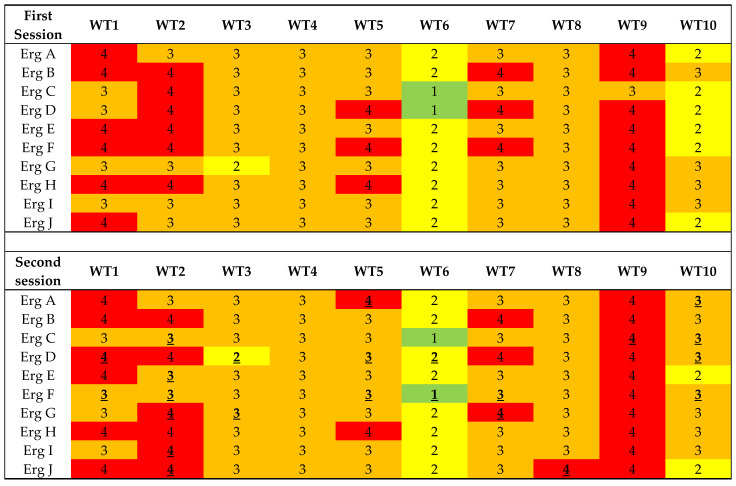
The results of the ten ergonomists (Erg A–J) that performed the risk level assessments using QEC (four levels) of the ten work tasks (WT) in both the first and second session. The colours refer to the different risk levels given in QEC. Green = Low, Yellow = Moderate, Orange = High, and Red = Very high. Bold and underlined numbers in the second session denote a difference from their first session result.

**Table 1 ijerph-20-05505-t001:** Descriptions of the ten different video-recorded work tasks in the study.

Work Task	Task Activity	Hours ^1^ per Workday	Handled Goods (kg)	Environment, Physical Factors	Discomfort (CR-10)	Work Demands and Control ^2^
1	Unpacking groceries to shelves in a supermarket store	just above 4	2	Good	3	Partly autonomy
2	Putting nets around roasts at a slaughterhouse	just above 4	2.5–4.5	Cold, wet, noisy	4	Group autonomy
3	Throwing small boxes into containers (post sorting)	just above 2	3	Cold during winter, warm during summer, noisy, difficulty concentrating	3–4	Controlled
4	Putting bundles of letters into boxes (post sorting)	approx. 6	2	Cold during winter, warm during summer, noisy, difficulty concentrating	3–4	Controlled
5	Deboning meat at a slaughterhouse	approx. 7	3–4	Cold, wet, noisy, sharp knives	3–4	Group autonomy
6	Assembling engines	just under 3	2	Good	2.5	Controlled
7	Cutting hair	just above 4	1	Good	3	Autonomy
8	Cleaning lavatories	approx. 5	1	Good	2	Partly autonomy
9	Supermarket cashier work	approx. 7	1–5	Good	3	Controlled
10	Cleaning stairs	just under 4	1	Usually good, sometimes cold	3	Partly autonomy

^1^ Pre-set task duration. ^2^ Autonomy: The worker controls the work himself/herself as if self-employed. Partly autonomy: The worker controls the work task but is limited in time and by obligations of other work tasks included in the work. Group autonomy: a group of employees control and divide work tasks within the group. Controlled: The work task is completely time-controlled by work instructions and space-controlled by the physical design of the workplace.

**Table 2 ijerph-20-05505-t002:** Description of body regions where posture and movements/repetitions were assessed from the video and rated by the ergonomists, for each of the six methods.

	Posture					Movement/Repetition				
	Back	Neck	Shoulder/ Arm	Elbow	Wrist/ Hand	Back	Neck	Shoulder /Arm	Elbow	Wrist/ Hand
**ART**	X	X	X		X			X		X
**HARM**		X	X		X		X	X	X	X
**OCRA**			X	X	X			X	X	X
**QEC**	X	X	X		X	X		X		X
**SI**					X					X
**SWEA**	X	X	X		X			X		X

**Table 3 ijerph-20-05505-t003:** Number of ergonomists that completed the risk assessments in the first session and in the second session (in brackets), number of items rated in each method, total number of performed item ratings and total number of risk level assessments, and the distribution, in percent, of risk levels in the assessments in the different risk levels stipulated in each of the methods, from “1” = lowest risk to “5” highest risk. The risk level distribution for the expert group is shown in bold print.

Method	Number of Ergonomists	Items Rated	Performed Ratings	Risk level Assessments	Distribution, in Percent, of Risk Levels ^1^ From Low (1) to High (5)				
					1	2	3	4	5
ART left arm	11 (9)	12	1320 (1080)	110 (90)	17 (17) **20**	44 (39) **50**	39 (44) **30**	-	-
ART right arm	11 (9)	12	1320 (1080)	110 (90)	8 (12) **10**	35 (32) **40**	56 (56) **50**	-	-
HARM	12 (8)	27	3240 (2160)	120 (80)	27 (25) **20**	48 (54) **70**	26 (21) **10**	-	-
OCRA	11 (10)	12	1320 (1200)	110 (100)	26 (28) **20**	15 (15) **20**	16 (19) **30**	33 (27) **20**	9 (11) **10**
QEC total	12 (10)	7	840 (700)	120 (100)	2 (2) **10**	15 (11) **0**	57 (59) **50**	27 (28) **40**	-
QEC Neck	12 (10)	2	240 (200)	120 (100)	0 (0) **0**	13 (14) **10**	54 (50) **60**	33 (36) **30**	-
QEC Shoulder	12 (10)	5	600 (500)	120 (100)	2 (2) **10**	38 (39) **20**	60 (59) **70**	0 (0) 0	-
QEC Wrist	12 (10)	5	600 (500)	120 (100)	0 (0) **0**	33 (35) **30**	68 (65) **70**	0 (0) 0	-
QEC Back	12 (10)	6	720 (600)	120 (100)	17 (14) **10**	38 (39) **30**	41 (44) **60**	5 (3) 0	-
SI highest score	12 (10)	6	720 (600)	120 (100)	3 (6) **20**	15 (6) **0**	83 (88) **80**	-	-
SWEA Overall Repetition	12 (8)	1	120 (80)	120 (80)	8 (11) **10**	42 (41) **30**	51 (48) **60**	-	-
SWEA Overall Postures and movements	12 (8)	1	120 (80)	120 (80)	20 (15) **20**	73 (70) **80**	8 (15) **0**	-	-
SWEA Neck posture ^2^	10 (6)	1	100 (60)	100 (60)	31 (13) -	55 (60) -	14 (27) -	-	-
SWEA Shoulder/arm Posture ^2^	10 (6)	1	100 (60)	100 (60)	26 (23) -	69 (72) -	5 (5) -	-	-
SWEA Back posture ^2^	10 (6)	1	100 (60)	100 (60)	22 (10) -	64 (82) -	14 (8) -	-	-

^1^ ART, HARM, SI, SWEA (three levels, 1–3); QEC (four levels 1–4); OCRA (five levels, 1–5). ^2^ No ratings were performed by the expert group.

**Table 4 ijerph-20-05505-t004:** Inter-observer reliability of risk levels, with *pre-given specific differing work task durations for the ten work tasks *(See Table 2). The item ratings were first used to compute the risk scores for each method and were then transposed into risk levels according to the instructions for each of the methods. Number of ergonomists (n), proportional agreement (%), Cohen’s kappa (K), linearly weighted kappa averaged over pairs (K_lw_), intraclass correlation (ICC) and Kendall’s coefficient of concordance (KCC) for the results of the ergonomists’ ratings of the ten different video-recorded work tasks.

			Inter-Observer Reliability				
Method	Assessment	N	%	K	K_lw_	ICC	KCC
ART	Left Arm (3 levels)	11	68	0.50	0.58	0.70	0.77
	Right Arm (3 levels)	11	78	0.59	0.65	0.75	0.72
HARM	Total (3 levels)	12	73	0.58	0.65	0.77	0.79
OCRA	Total (5 levels)	11	39	0.21	0.43	0.62	0.65
QEC	Total (4 levels)	12	68	0.46	0.55	0.69	0.72
	Neck (4 levels)	12	91	0.85	0.87	0.90	0.92
	Shoulder (4 levels)	12	71	0.42	0.44	0.51	0.57
	Wrist (4 levels)	12	86	0.67	0.67	0.70	0.73
	Back (4 levels)	12	57	0.35	0.49	0.67	0.70
SI ^1^	Highest score (3 levels)	12	83	0.20	0.18	0.18	0.33
SWEA	Overall repetition (3 levels)	12	58	0.26	0.30	0.39	0.48
	Overall postures and movements (3 levels)	12	65	0.18	0.21	0.28	0.35
	Neck posture (3 levels)	10	51	0.17	0.22	0.32	0.45
	Shoulder/arm posture (3 levels)	10	56	0.16	0.21	0.29	0.40
	Back posture (3 levels)	10	60	0.12	0.16	0.25	0.34

^1^ Although instructed to rate both hands, the ergonomists sometimes did not rate the less active hand, therefore in each video the hand that yielded the highest score was used for the inter-observer reliability computations.

**Table 5 ijerph-20-05505-t005:** Inter-observer reliability of risk levels, *with standardized work task duration, using 3 h 45 min for all ten work tasks*. The item ratings were first used to compute the risk scores for each method and were then transposed into risk levels according to the instructions for each of the methods. Number of ergonomists (n), proportional agreement (%), Cohen’s kappa (K), linearly weighted kappa averaged over pairs (K_lw_), intraclass correlation (ICC) and Kendall’s coefficient of concordance (KCC) for the results of the ergonomists’ ratings of the ten different video-recorded work tasks.

			Inter-Observer Reliability				
Method	Assessment	N	%	K	K_lw_	ICC	KCC
ART	Left Arm (3 levels)	11	60	0.32	0.42	0.57	0.66
	Right Arm (3 levels)	11	62	0.34	0.41	0.55	0.56
HARM	Total (3 levels)	12	75	0.26	0.26	0.30	0.36
OCRA	Total (5 levels)	11	42	0.25	0.45	0.64	0.71
QEC	Total (4 levels)	12	76	0.39	0.42	0.51	0.56
	Neck (4 levels)	12	91	0.82	0.82	0.84	0.86
	Shoulder (4 levels)	12	63	0.29	0.33	0.44	0.52
	Wrist (4 levels)	12	78	0.48	0.48	0.51	0.55
	Back (4 levels)	12	59	0.24	0.30	0.40	0.50
SI ^1^	Highest score (3 levels)	12	81	0.16	0.15	0.14	0.29
SWEA ^2^	Overall repetition (3 levels)	12	58	0.26	0.30	0.39	0.48
	Overall postures and movements (3 levels)	12	65	0.18	0.21	0.28	0.35
	Neck posture (3 levels)	10	51	0.17	0.22	0.32	0.45
	Shoulder/arm posture (3 levels)	10	56	0.16	0.21	0.29	0.40
	Back posture (3 levels)	10	60	0.12	0.16	0.25	0.34

^1^ Although instructed to rate both hands, the ergonomists sometimes did not rate the less active hand, therefore in each video the hand that yielded the highest score was used for the inter-observer reliability computations. ^2^ For SWEA, the work task duration is not part of the methods, hence the results with and without standardised work task duration is the same.

**Table 6 ijerph-20-05505-t006:** The inter-observer reliability, in the linearly weighted Kappa (K_lw_), for the items in each method that were rated by the ergonomists (in the first session) from the video-recorded work tasks. The item with the lowest K_lw_ (Min item), and the item with the highest K_lw_ (Max item) refer to the min and max columns.

Method	K_lw_ Min–Max	Min Item	Max Item
ART	0.17–0.44	Wrist posture	Arm/hand repetition
HARM	0.14–0.30	Forearm/wrist posture	Force exertions
OCRA	0.03–0.53	Elbow movement	Repetitiveness
QEC	0.17–0.44	Hand/wrist posture	Hand/wrist movements
SI ^1^	0.17–0.42	Hand/wrist posture	Efforts per minute
SWEA	0.16–0.22	Back posture	Neck posture

^1^ Although instructed to rate both hands, the ergonomists sometimes did not rate the less active hand, therefore in each video the hand that the highest number of ergonomists had rated (i.e., the right hand in all tasks but meat netting) was used for the reliability computations for the rated items.

**Table 7 ijerph-20-05505-t007:** Intra-observer reliability of risk levels, with *pre-given specific differing work task durations for the ten work tasks *(See Table 2). Number of ergonomists (n), proportional agreement (%) mean Cohens kappa (K), linearly weighted kappa averaged over pairs (K_lw_), intraclass correlation (ICC) and Kendall’s coefficient of concordance (KCC) for the results of the ergonomists’ ratings of the ten different video-recorded work tasks. The ratings were first used to compute the score for each method and then converted into risk levels.

			Intra-Observer Reliability				
Method	Assessment	n	%	K	K_lw_	ICC	KCC
ART	Left Arm (3 levels)	9	74	0.59	0.65	0.74	0.88
	Right Arm (3 levels)	9	79	0.62	0.68	0.78	0.86
HARM	Total (3 levels)	10	78	0.64	0.70	0.79	0.89
OCRA	Total (5 levels)	10	45	0.29	0.52	0.72	0.85
QEC	Total (4 levels)	10	77	0.60	0.68	0.79	0.88
	Neck (4 levels)	10	92	0.87	0.88	0.92	0.96
	Shoulder (4 levels)	10	78	0.57	0.58	0.62	0.83
	Wrist (4 levels)	10	89	0.76	0.76	0.77	0.89
	Back (4 levels)	10	67	0.49	0.60	0.74	0.87
SI ^1^	Highest score (3 levels)	10	77	0.15	0.13	0.10	0.56
SWEA	Overall repetition (3 levels)	8	68	0.41	0.47	0.56	0.80
	Overall postures and movements (3 levels)	8	71	0.27	0.30	0.36	0.68
	Neck posture (3 levels)	6	62	0.24	0.32	0.47	0.76
	Shoulder/arm posture (3 levels)	6	67	0.09	0.13	0.20	0.60
	Back posture (3 levels)	6	72	0.41	0.44	0.51	0.75

^1^ Although instructed to rate both hands, the ergonomists sometimes did not rate the less active hand, therefore in each video the hand that yielded the highest score was used for the intra-observer reliability computations.

**Table 8 ijerph-20-05505-t008:** Intra-observer reliability of risk levels, *with standardized work task duration, using 3 h 45 min for all ten work tasks*. Number of ergonomists (n), proportional agreement (%) mean Cohens kappa (K), linearly weighted kappa averaged over pairs (K_lw_), intraclass correlation (ICC) and Kendall’s coefficient of concordance (KCC) for the results of the ergonomists’ ratings of the ten different video-recorded work tasks. The ratings were first used to compute the score for each method and then converted into risk levels.

			Intra-Observer Reliability				
Method	Assessment	n	%	K	K_lw_	ICC	KCC
ART	Left Arm (3 levels)	9	72	0.50	0.56	0.66	0.84
	Right Arm (3 levels)	9	71	0.49	0.56	0.66	0.82
HARM	Total (3 levels)	10	81	0.46	0.47	0.21	0.71
OCRA	Total (5 levels)	10	48	0.31	0.50	0.68	0.83
QEC	Total (4 levels)	10	81	0.56	0.58	0.62	0.81
	Neck (4 levels)	10	92	0.84	0.84	0.85	0.93
	Shoulder (4 levels)	10	69	0.39	0.42	0.49	0.75
	Wrist (4 levels)	10	84	0.63	0.64	0.67	0.83
	Back (4 levels)	10	73	0.44	0.48	0.57	0.80
SI ^1^	Highest score (3 levels)	10	73	0.22	0.16	0.12	0.58
SWEA^2^	Overall repetition (3 levels)	8	68	0.41	0.47	0.56	0.80
	Over all postures and movements (3 levels)	8	71	0.27	0.30	0.36	0.68
	Neck posture (3 levels)	6	62	0.24	0.32	0.47	0.76
	Shoulder/arm posture (3 levels)	6	67	0.09	0.13	0.20	0.60
	Back posture (3 levels)	6	72	0.41	0.44	0.51	0.75

^1^ Although instructed to rate both hands, the ergonomists sometimes did not rate the less active hand, therefore in each video the hand that yielded the highest score was used for the intra-observer reliability computations. ^2^ For SWEA, the work task duration is not part of the methods, hence the results with and without standardised work task duration is the same.

**Table 9 ijerph-20-05505-t009:** Concurrent validity of risk levels *with standardized work task duration, using 3 h 45 min for all ten work tasks*. Number of ergonomists (n), proportional agreement (%), Cohen’s kappa (K), linearly weighted kappa averaged over pairs (K_lw_), intraclass correlation (ICC) and Kendall’s coefficient of concordance (KCC) for the results of the ergonomists’ ratings of the ten different video-recorded work tasks. The ratings were first used to compute the score for each method and then converted into risk levels. To assess the validity, the ergonomists’ ratings were compared to those of a group of three experts.

			Concurrent validity				
Method	Assessment	n	%	K	K_lw_	ICC	KCC
ART	Left Arm (3 levels)	11	66	0.46	0.54	0.65	0.84
	Right Arm (3 levels)	11	65	0.41	0.48	0.60	0.78
HARM	Total (3 levels)	12	74	0.41	0.42	0.44	0.73
OCRA	Total (5 levels)	11	44	0.28	0.44	0.59	0.79
QEC	Total (4 levels)	12	75	0.35	0.47	0.63	0.81
	Neck (4 levels)	12	94	0.88	0.88	0.89	0,95
	Shoulder (4 levels)	12	67	0.43	0.48	0.58	0.81
	Wrist (4 levels)	12	84	0.58	0.58	0.61	0.81
	Back (4 levels)	12	68	0.34	0.31	0.29	0.66
SI ^1^	Highest score (3 levels)	12	83	0.34	0.35	0.36	0.69
SWEA	Overall repetition (3 levels)	12	59	0.27	0.31	0.38	0.69
	Overall postures and movements (3 levels)	12	76	0.36	0.38	0.44	0.73

^1^ Although instructed to rate both hands, the ergonomists sometimes did not rate the less active hand, therefore in each video the hand that yielded the highest score was used for the intra-observer reliability computations.

## Data Availability

Data available on request due to ethical restrictions. The data presented in this study are available on request from the corresponding author. The data are not publicly available since the informants in the study have been guaranteed anonymity.

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
