# Peer review of "Reliability and Validity of Six Selected Observational Methods for Risk Assessment of Hand Intensive and Repetitive Work"

_ijerph, 2023, doi:10.3390/ijerph20085505_

Round 1
Reviewer 1 Report
this is good work. You may wish to consider reducing the discussion section as much of the information is presented within the results. There are elements in the results that should be in method

Author Response
We would like to thank the reviewer for valuable comments and for taking the time to review the manuscript. Please see the attachment for the authors' response to the reviewer.
Kind regards,
Teresia Nyman (on behalf of the authors)

Reviewer 2 Report
The paper addresses an important problem in risk assessment, which is the reliability of information.
The study is well prepared and documented. The results are supported by the research conducted. The used literature a good description of the research environment.
Further development activities were indicated.
The authors have put a lot of effort into the development. It should be recognized that the presented scope of work authorizes its publication.
The only caveat are minor typos, e.g. in literature sources.
Author Response
We wish to thank the reviewer for very positive response, and for taking the time to review the manuscript. We will check the bibliography for spelling errors.
Kind regards,
Teresia Nyman (on behalf of all authors)
Reviewer 3 Report
The topic is interesting and indeed linked to a major problem in our society - the assessment of human health when performing various occupational activities involving hand/wrist postures.
The authors stated that they analysed and evaluated six risk assessment methods for inter- and intra-observer reliability and concurrent validity using the same methodological design and statistical parameters. This study involved twelve ergonomics specialists, people with experience in the field.
I kindly ask to clarify the following details:
1. in the abstract, the authors said "twelve experienced ergonomists per-formed risk assessments of the ten video-recorded work tasks twice and consensus assessments for the concurrent validity were carried out by three experts. ".... In the section named Results....lines 332-337 it is said" in the first session, at least 10 out of 12 recruited ergonomists completed the assessments of all items in all methods.....In the second session, at least six out of 12 recruited ergonomists completed the assessments using all methods (Table 3)" Perhaps it is necessary to justify this detail or explain why it was done this way.
2.it is said that the ergonomists were trained to use all the methods selected for this study. They applied the selected methods (after a short training using video materials ) to assess the risk for different activities. The question is: Was it sufficient to assess the mentioned activities with video material and/or supplementary written material they received without making personal observations?
3. the experts used selected methods to evaluate body movements and repetitions (see Table2, 3, ...). The authors have analysed the experts' results, but perhaps it was useful to know which method the experts recommend for assessing a specific physical activity (depending on their expertise and professional background).
Author Response
We would like to thank the reviewer for valuable comments and for taking time to review our manuscript.
Our answers to the reviewer's comments are uploaded in the enclosed file.
Kind regards,
Teresia Nyman (on behalf of the authors)

Reviewer 4 Report
A Well written manuscript acceptable in the current form
Author Response
We wish to thank the reviewer for very positive response, and for taking the time to review the manuscript.
Kind regards,
Teresia Nyman (on behalf of all authors)